# Recurrence of Head and Neck Squamous Cell Carcinoma: Did the COVID-19 Pandemic Have an Impact on Therapeutic Management?

**DOI:** 10.3390/jcm14207406

**Published:** 2025-10-20

**Authors:** Benjamin Reliquet, Thomas Thibault, Paul Elhomsy, Dounia Chbihi, Mireille Folia, Caroline Guigou

**Affiliations:** 1Department of Otolaryngology-Head and Neck Surgery, CHU Dijon Bourgogne, Université Bourgogne Europe, 21000 Dijon, France; benjamin.reliquet@chu-dijon.fr (B.R.); mireille.folia@chu-dijon.fr (M.F.); 2Department of Internal Medicine, CHU Dijon Bourgogne, Université Bourgogne Europe, 21000 Dijon, France; thomas.thibault@chu-dijon.fr; 3Anesthesiology and Critical Care Department, CHU Dijon Bourgogne, Université Bourgogne Europe, 21000 Dijon, France; paul.elhosmy@chu-dijon.fr; 4Department of Otolaryngology-Head and Neck Surgery, Croix Rousse Hospital, 69004 Lyon, France; 5ICMUB Laboratory, UMR CNRS 6302, Department of Otolaryngology-Head and Neck Surgery, CHU Dijon Bourgogne, Université Bourgogne Europe, 21000 Dijon, France

**Keywords:** recurrent squamous cell carcinoma, COVID-19 pandemic, upper aerodigestive tract, professional practices, multidisciplinary team meetings

## Abstract

**Objectives**: To our knowledge, no previous study has specifically investigated the impact of COVID-19 on the management of recurrent squamous cell carcinoma (HNSCC) of the upper aerodigestive tract. The aim was to investigate the impact of the COVID-19 pandemic on the adequacy of treatments decided upon in multidisciplinary team meetings (MTMs) and those actually administered to patients with recurrent tumors. Secondary objectives were to study the characteristics of this population and tumors, factors that may influence the mismatch between treatments decided upon and those administered, and treatment delays. **Methods**: A retrospective study conducted at a tertiary referral center included 80 patients with recurrent HNSCC diagnosed between 2019 and 2021. **Results**: Of the 80 patients, 19 (24%) received treatment that differed from the recommended treatment. In case of mismatching, treatments were mostly less invasive or palliative (32% of palliative treatments decides (n = 6) versus 64% of palliative treatments realized (n = 12)). No factors were identified that could explain the discrepancy between treatment decided and treatment administered. The study population was homogeneous over the 3 years, with the only difference being lower lymph node staging in 2020 (*p* = 0.002). The time to recurrence was longer (*p* < 0.001), and the time to treatment initiation was shorter during the pandemic and post-pandemic periods (*p* = 0.002). **Conclusions**: The COVID-19 pandemic did not appear to affect the consistency between treatment decisions made in MTM and the treatments ultimately delivered. The COVID-19 pandemic has enabled earlier treatment with less advanced staging in patients with recurrence.

## 1. Introduction

Head and neck squamous cell carcinoma (HSNCC) ranks as the sixth most diagnosed cancer worldwide [1]. It is estimated that 15 to 50% of HSNCC cases will recur [2,3,4].

Tumor recurrence is defined as the re-emergence of a tumor exceeding 6 months following the completion of initial treatment [5]. Tumor recurrence can manifest as local, lymph node (regional), or distant sites [6]. The treatment modalities for tumor recurrence differ from those used for the initial diagnosis [6]. Surgery is more complex due to tissue alterations induced by prior treatment [7], and radiotherapy must consider the possibility of overdose on sensitive organs [8]. The development of immunotherapy and targeted therapies offers systemic alternatives [9]. Even with multimodal treatment, the prognosis remains poor, with a median survival of between 6 and 15 months [10,11].

Patients experiencing tumor recurrences should be distinguished from patients with a second tumor because of the different prognoses [12,13]. Similarly, patients with progressive disease, defined as oncological progression less than 6 months after the completion of curative treatment, require differentiation from those with true recurrence [14].

Similarly to patients receiving an initial cancer diagnosis, all cases of tumor recurrence are presented at a multidisciplinary team meeting (MTM). This is an essential and mandatory step in France (French Public Health Code, Article D. 6124-131). The multidisciplinary team meeting marks the end of the diagnostic phase and the start of the therapeutic phase [15]. Given its inherent multidisciplinary nature, it allows for consensus to be reached between the various specialties present and the recommendations in force, thereby enabling the establishment of a personalized care plan [16].

The impact of the COVID-19 pandemic on oncological or non-oncological cervico-facial surgery in France has been explored in several prior studies [17,18,19]. Like everyone else, ENT surgeons have had to adapt their care to the pandemic context [20].

In a previous study [21], the impact of COVID-19 pandemic was examined on the adequacy between the treatment decided upon in MTM and the treatment actually provided. This study included 475 patients, revealing that the consistency rate between MTM decisions and administered treatment remained stable at approximately 73%, irrespective of the pre-pandemic, peri-pandemic, or post-pandemic periods. This study aligns with to data in the literature, which ranged from 73% to 85% [22,23,24]. The COVID-19 pandemic influenced the characteristics of the patients and tumors managed during this period. Specifically, during 2020 (the COVID-19 pandemic year), patients presented with significantly higher rates of undernourishment (*p* < 0.001) and had a more impaired Performance Status (PS ≥ 2) (*p* < 0.001). The rate of disease detection at a metastatic stage increased from 4.8% in 2019 to 12% in 2020, suggesting potential barriers to accessing ENT consultations. In addition, the time between the first specialist consultation and MTM (diagnostic time) remained stable, while the time between MTM and the first day of treatment (therapeutic time) was shortened (32 days in 2019 versus 24 days in 2020 and 26 days in 2021, *p* < 0.001). This suggests an adaptive capacity of the healthcare system during the pandemic, allowing for the maintenance of adequate patient care within tertiary referral centers. Long-term results were also reassuring, notably due to the absence of an increase in the rate of newly diagnosed cancer and the absence of diagnoses at more advanced stages in the years following the COVID-19 pandemic in patients with HNSCC [25,26].

To our knowledge, studies in the scientific literature dealing with this subject have focused only on newly diagnosed patients [21,24,25]. No previous study has specifically investigated the impact of COVID-19 on the management of recurrent cases. Patients with tumor recurrence have specific characteristics that do not allow the results for newly diagnosed patients to be extrapolated [10]. We therefore felt it was important to look into this subject, which is rarely addressed in scientific literature.

The primary objective was to assess the impact of the COVID-19 pandemic on the appropriateness of decisions made in MTMs and treatments administered for recurrent HNSCC. The secondary objectives were to study the characteristics of this population and the tumors, to identify factors that may influence the mismatch between the treatments decided upon in the multidisciplinary team meeting and those actually performed, and to study the time taken to initiate treatment.

## 2. Materials and Methods

### 2.1. Study Design

This is a retrospective, descriptive, and single-center study conducted in a tertiary referral hospital from 1 January 2019 to 31 December 2021.

2019 served as the pre-COVID-19 reference year and has already been covered in previous work [27]. The year 2020 was seen as the pandemic period. The year 2021 was considered the post-pandemic period after the COVID-19 vaccine came out. The first vaccine hit the market on 17 December 2020 [28]. From that point on, the COVID-19 pandemic was gradually seen as “under control.”

The patients’ data were anonymised, and managed in conformity with the MR-004 reference methodology of the National Commission on Informatics and Liberty (CNIL). All patients received a letter informing them of the collection of data from their medical records and requesting their non-objection. This study was conducted in lines with the rules of good clinical practice, and the requirement for the Ethics Committee approval was not required. The study was reviewed by the Committees for the Protection of Persons (CPP Est I), which determined that it did not fall under the scope of the Jardé law. This study was recorded under the Health Data Hub number: 20646009.

The inclusion criteria were:-Files recorded and discussed in Multidisciplinary Team meetings (MTM) at the University Hospital Centre between 1 January 2019 and 31 December 2021.-Prior diagnosis of HNSCC of the oral cavity, oropharynx, hypopharynx, or larynx.-Completion of prior treatment for the aforementioned carcinoma at least 6 months prior to the study period.-Availability of both planned and executed treatment plans in their records

The exclusion criteria were as follows:-First-time diagnosis of HNSCC diagnoses.-Tumor localization in the salivary glands, sinuses, or nasal cavities.-Minor patients (under 18 years of age).-Cutaneous squamous cell carcinoma.-Histological findings other than squamous cell carcinoma.-Files presented in MTM solely for expert discussion.-Unavailable treatment plans.

### 2.2. Data Collected

The Cancer Coordination Center the University Hospital Centre provided the registry of patients enrolled in MTMs. A range of clinical and demographic variables were collected from the patient’s medical records:-Patients’ specifications and medical history including gender, age, World Health Organization Performance Status (PS) index [29], cirrhosis, diabetes, chronic obstructive pulmonary disease (COPD), history of malnutrition, cardiovascular history, history of other cancers (current or remission), and past and/or current alcohol and tobacco use.-HNSCC characteristics: Site (oral cavity, oropharynx, hypopharynx, larynx, or cervical lymphadenopathy without an identified primary tumor site), Tumor Node Metastasis (TNM) status at time of diagnosis presented according to the eighth edition of the UICC (Union for International Cancer Control) 2017 TNM classification [30], histopathology, and human papillomavirus (HPV) status.-Initial Extension Assessment According to the Recommendations of the French Society of Otorhinolaryngology (SFORL) which included: pan endoscopy of the upper aerodigestive tract with the provision of a summary diagram and an operative report; Ear, Nose, and Throat (ENT) MRI; cervical-thoracic (CT) scan; Positron Emission Tomography (PET-CT scan); or another ultrasound/scan-guided biopsy.-Multidisciplinary team meeting (MTM): These meetings were held weekly and jointly by the University Hospital Center and the affiliated cancer center. Each meeting involved at least 3 medical or surgical specialties from among the following: ENT, maxillofacial or reconstructive surgeon, medical oncologists, radiation oncologists, radiologists, and pathologists. Before each meeting, the patient’s referring physician completed and verified a standardized form. During the meeting, the patient’s case was described, and each aspect of the extension assessment was analyzed. The form included the previous relevant data, attending physicians, and the date. In our center, the patient is not physically present at the MTM. After the meeting, the MTM coordinator summarized collective decision on the treatment protocol and validated it. The document summarizing the discussions and the MTM recommendation was placed in the patient’s electronic medical record. The collective treatment protocol decision was shared with the patient during the consultation conducted by the referring ENT physician after the meeting. During this consultation, the patient provided consent or refusal regarding the proposed treatment.-Various time intervals were calculated from the patient’s medical record: the diagnostic delay (time between the consultation that suspected recurrence and the MTM finalizing the treatment), and the time to treatment initiation (time between the MTM and the start of treatment).-History during the remission duration defined by the date of completion of the first treatment and the consultation with a specialist confirming the recurrence: time of remission, prior treatment (surgery, radiotherapy, chemotherapy).-Treatment center was registered (reference center or other center), and 3-year survival.

### 2.3. Statistical Analysis

All data tables were compiled using Microsoft Excel (version 2022, Redmond, WA, USA). R software (version 4.1.2., Miami, FL, USA) was used to conduct the statistical analyses.

Patient characteristics, tumor features, extension assessments, and treatment modalities are expressed as percentages for qualitative variables. Quantitative variables were expressed as median (interquartile range). Bivariate data comparison was used to compare categorical variables. Fisher’s exact test was employed for qualitative variables, and the Wilcoxon–Mann–Whitney or Student’s *t*-test was employed for quantitative variables according to distribution. The explanatory variables used were (1) tumor features, (2) patient characteristics, and (3) extension assessments. These variables were used to compare the 2 groups established by the alignment or nonalignment between the treatment received and the treatment decided in MTMs (MTM—Treatment Matching). Multivariate analysis was performed using logistic regression with the outcome variable “MTM—Treatment Matching” to adjust for potential confounders. Covariates with a *p*-value less than 0.2 in the bivariate analysis were used in the multivariate model [31]. We performed a multivariate analysis with a limited number of predictors to avoid overfitting [32]. Multicategorical variables (those with more than 2 categories) were dichotomized to meet the log-linearity assumption. The *p*-value of less than 0.05 was evaluated as significant.

## 3. Results

### 3.1. Population Characteristics

Between 1 January 2019 and 31 December 2021, 1202 patient medical records were presented at MTM. Of these, 80 patients met all the inclusion criteria and were consequently enrolled in the study.

The flowchart is detailed in Figure 1.

The characteristics of the included population are summarized in Table 1.

The study population’s demographic and clinical characteristics were consistent across the 3 periods studied. The cohort’s vast majority were male (n = 68; 85%) with a median age of 67. A quarter (n = 20) of our population had a PS score ≥ 2. There were no differences in the characteristics of the population when comparing 2019 to 2020 and 2019 to 2021.

The characteristics of the initial tumors are summarized in Table 2.

A significant difference was observed in the distribution of primary tumor sites over time. Notably, hypopharyngeal tumors accounted for 31% (n = 10) of cases in 2019 but dropped to only one case in 2020 and were absent in 2021. Conversely, oropharyngeal tumors increased in frequency.

Patients with tumor recurrence treated before the COVID-19 pandemic appeared to have more advanced lymph node staging than those treated during or after the COVID-19 pandemic (16 N + patients (50%) in 2019, versus 2 patients (7%) in 2020, and 5 patients (25%) in 2021, *p* = 0.002).

The distribution of initial treatment modalities also varied significantly by year. In 2019 (before COVID-19), the most common treatment was radiochemotherapy (RTCT, n = 15, 47%), whereas in 2020, surgery alone became the predominant approach (n = 13, 46%). By 2021, combined modalities such as surgery followed by RTCT became more frequent (n = 5, 25%).

### 3.2. Treatment Delays

The different processing times and circumstances of discovery are summarized in Table 3.

The time to tumor recurrence appears to be longer during the “pandemic” and “post-pandemic” periods (22 months in 2019, versus 63 months in 2020 and 48 months in 2021, *p* < 0.001).

The time between MTM’s decision and the start of treatment also appears to have shortened during these two periods (30 days in 2029 versus 16 days in 2020 and 24 days in 2021, *p* = 0.002) regardless of the treatment provided.

### 3.3. Treatment Performed

The treatments performed each year are summarized in Table 4.

### 3.4. Mismatch Between Treatment Prescribed and Treatment Performed

In the population, 19 patients (24%) had treatment different from that decided in the multidisciplinary team meeting (proportion of mismatched cases): 6 patients (19%) in 2019, 6 patients (21%) in 2020, and 7 patients (35%) in 2021. This difference is not significant between years (*p* = 0.4, Wilcoxon test).

Table 5 shows the characteristics of populations presenting a mismatch or match between the treatment decided upon in the MTM and the treatment actually administered.

During multivariate analysis; no population factors were found that could contribute to the mismatch between the treatment decided upon in MTM and the treatment actually administered (Table 6).

Figure 2 shows the treatments performed for the 19 patients for whom the treatment decided upon in the multidisciplinary team meeting was not carried out.

In cases of mismatching, the treatment performed was less invasive or even palliative (32% of palliative treatments decided upon (n = 6) versus 64% of palliative treatments performed (n = 12)). For 47% of patients (n = 9), no treatment was performed, regardless of whether it was in 2019, 2020, or 2021. For 32% (n = 6) of these patients, palliative chemotherapy was decided upon in the multidisciplinary team meeting. For half of the patients eligible for surgery (n = 2), no treatment was ultimately performed.

## 4. Discussion

This study aimed to observe the impact of the COVID-19 pandemic on the management of recurrent head and neck cancer. Specifically, we assessed the concordance between MTM recommendations and the treatment actually administered, and identified factors potentially influencing any observed discordance. Our findings revealed that no increase in treatment mismatch was observed in patients with recurrent HSNCC due to the COVID-19 pandemic. The observed discordance rate (19%) was within the range reported for newly diagnosed populations (5–23%) [21,22,23,24]. No patient or tumor characteristics were identified that could explain this mismatch. However, the non-significant but numerically higher non-concordance rate in 2021 (35%) could be a delayed effect of healthcare reorganisation or patient-related factors. In cases of treatment mismatch, the majority of patients received palliative care or no treatment at all (n = 11, 59%). The integrity of decisions regarding therapeutic drug management was preserved, even in complex and recurrent cases, a particularly vulnerable group during the pandemic. These results demonstrate the value of systemic treatments for this patient population. It is important to consider this during multidisciplinary team meetings.

The characteristics of the population with tumor recurrence were identical between the 3 periods. However, there was a change in lymph node staging, with fewer locally advanced tumors detected during the pandemic and post-pandemic periods (*p* = 0.002). Tumor recurrence times were longer (*p* = 0.002) with a delay in treatment during the pandemic and post-pandemic periods (*p* < 0.001). This finding aligns with our previous study [21], which demonstrated a shortened time to treatment once patients entered the healthcare system, thereby indicating the system’s adaptive capacity and sustained functionality during the pandemic.

Recurrent tumors and progressive disease represent a different challenge than newly diagnosed patients: changes in anatomy, surgical or radiation history alter the possibilities for new therapy [7,33]. Advances in surgical techniques are an important issue, particularly in minimizing extensive surgery on frail patients who have undergone surgery or radiation therapy. Thus, the development of minimally invasive approaches appears to promise both satisfactory tumor control and an acceptable quality of life [34]. The emergence of targeted therapies and immunotherapy, which, together with recent advances, offer new therapeutic possibilities [14,35]. Some alternatives, such as proton therapy, are not always used in routine practice due to lack of accessibility [36].

Our study is part of a study of MTM practice, which remains an essential and mandatory routine in medical and surgical oncology. Beyond the legal framework, it allows for the comparison of perspectives from different but complementary medical specialties [37]. A persistent challenge, however, lies in the final treatment decision, which necessitates shared decision-making with the patient. MTMs must be improved to take into account patients as a whole. In our institution, patients do not directly participate in MTM meetings. This may be an area for improvement, even though a recent national study in our specialty showed no difference [38]. Digital or virtual MTMs could also be developed to enable remote patient participation. The development of artificial intelligence could also improve MTM decisions in the coming years by comparing them to the guidelines issued by learned societies.

The 3-year survival rate in this study is 34%. This rate is consistent with that reported in the literature. A meta-analysis on salvage surgery for recurrent head and neck cancer reported overall survival at 5-year rates ranging from 26% to 67% for surgical interventions versus 0% to 32% for non-surgical management [7]. The prognosis for patients with tumor recurrence is poorer than for patients with a primary diagnosis. However, certain factors have been considered to be good prognostic indicators in the literature. For example, salvage surgery, whether or not combined with radiotherapy or chemotherapy, has a better prognosis than other therapeutic modalities [10]. Temporal factors have also been highlighted: the longer it takes for recurrence to occur, the better the prognosis (with a cutoff point of 1 year for Spencer et al.) [6,12].

Follow-up is essential in the context of secondary prevention [39]. In our study, 36% of recurrences were discovered during clinical examination. This finding supports the recommendations from professional societies [40] on the regularity of follow-up, which may be joint and alternated with radiotherapists where appropriate. A routine follow-up consultation for a treated patient triggers a diagnosis of recurrence in one out of 34 consultations, whereas when the patient initiates an early consultation, the diagnosis rate rises to one in three consultations [41].

Our study has some limitations. First, it is a retrospective, single-center study. However, thanks to the method used to collect medical records and the standardization of information sheets during MTM, no major data were missing that could have altered our analysis. Nevertheless, it would be interesting to conduct a national multicenter analysis in order to compare the results of different French centers, in particular, response times during the pandemic. In this study, treatment initiation times were examined regardless of the treatment delivered. It would be interesting to examine treatment initiation times according to the type of treatment (surgery, radiotherapy, or systemic treatment). Another important limitation is the relatively small sample size, which results in a lack of statistical power. It would also be interesting to know the reasons for the change in treatment or even the absence of treatment (deterioration in the patient’s general condition, rapid tumor growth, patient refusal of treatment). With regard to the generalizability of the results, it is important to note the well-structured French MTM system. The results might be more robust than in other healthcare settings. Finally, it would have been relevant to know whether the COVID-19 pandemic caused a delay in the implementation of adjuvant post-surgical radio-(chemo)therapy. This is associated with a higher recurrence or continuation rate, regardless of the various factors discussed here [42].

## 5. Conclusions

This study looked at the consequences of the COVID-19 pandemic on the management of recurrent squamous cell carcinomas of the upper aerodigestive tract, particularly the adequacy of treatments decided upon in multidisciplinary team meetings compared to those actually performed. No increase in mismatches was observed during or after the pandemic, and no significant individual variables could be identified to explain these mismatches.

Despite the disrupted healthcare environment, the time taken to start treatment actually shortened, demonstrating the effective adaptation of the healthcare system. Regular follow-up remains a key element in the early diagnosis of recurrences. Finally, this study highlights the central role of multidisciplinary team meetings in treatment decisions, while paving the way for consideration of more active patient involvement in these decisions.

Larger-scale multicenter studies would be needed to confirm these observations and explore in greater detail the causes of discrepancies between decisions and treatment actions.

## Figures and Tables

**Figure 1 jcm-14-07406-f001:**
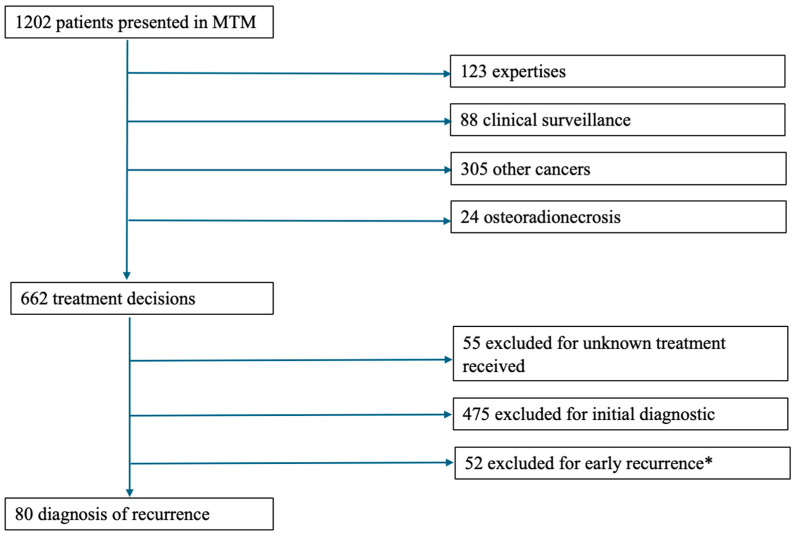
Flowchart. MTM: multidisciplinary team meeting. *: defined as tumor recurrence less than 6 months after the end of curative treatment [14].

**Figure 2 jcm-14-07406-f002:**
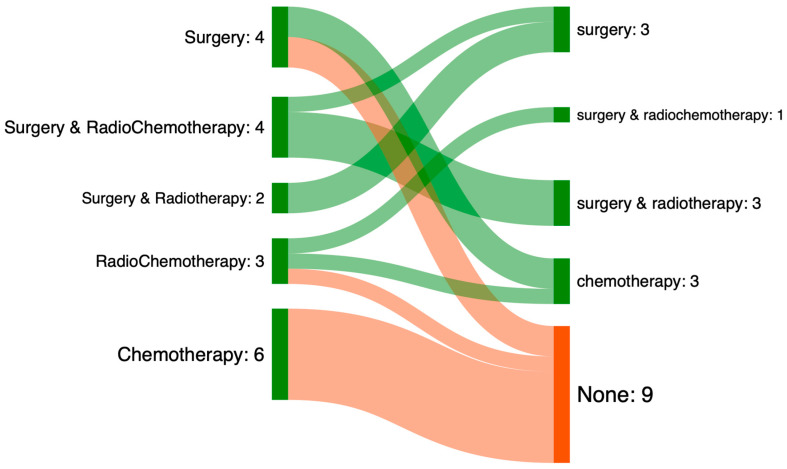
Sankey diagram showing treatments performed in cases of non-compliance with treatments decided upon in MTM. The left column shows the treatments decided upon, the right column shows the treatments performed.

**Table 1 jcm-14-07406-t001:** Characteristics of the population. COPD: chronic obstructive pulmonary disease.

	2019 N = 32 (100%)	2020 N = 28 (100%)	2021 N = 20 (100%)	*p*-Value ^2^
**Age** (years) ^1^	65	68	68	0.5
**Gender (male)**	28 (88)	23 (82)	17 (85)	>0.9
**Performance Status**				0.056
0	9 (28)	8 (29)	2 (10)	
1	20 (62)	11 (39)	11 (55)	
2	2 (6)	5 (18)	5 (25)	
3	3 (10)	2 (7)	1 (5)	
4	0	1 (4)	1 (5)	
**Medical history:**				
COPD	10 (31)	5 (18)	1 (5)	0.068
Hepatopathy	4 (12)	1 (4)	0 (0)	0.2
Cardiovascular disease	20 (63)	14 (50)	11 (55)	0.6
Diabetis	7 (21)	7 (25)	6 (30)	0.8
Undernutrition	18 (56)	23 (82)	12 (60)	0.090
**Toxic habit:**				
Active smoking	72 (34)	7 (25)	5 (25)	0.8
Pack-Years (PY) ^1^	41	43	40	
Alcoholism	5 (16)	11 (39)	5 (25)	0.6
3 years survival	11 (32)	8 (28)	8 (40)	0.8

^1^ Age and number of pack-years are expressed as averages; ^2^ Kruskal–Wallis rank sum test; Fisher’s exact test.

**Table 2 jcm-14-07406-t002:** Characteristics of the initial tumors. RT: radiotherapy. RTCT: radiochemotherapy.

	2019 N = 32 (100%)	2020 N = 28 (100%)	2021 N = 20 (100%)	*p*-Value
**Primary Tumor Site**				<0.001
Oral Cavity	12 (38)	23 (43)	6 (30)	
Oropharynx	5 (16)	9 (32)	8 (40)	
Larynx	5 (16)	6 (21)	6 (30)	
Hypopharynx	10 (31)	1 (4)	0	
**HPV 16 positive status**	2 (6)	1 (4)	4 (20)	0.13
**Initial staging**				
T1/T2	13 (41)	15 (54)	11 (55)	0.5
T3/T4	19 (59)	13 (46)	9 (45)	
N+	16 (50)	2 (7)	5 (25)	0.02
M+	0	1 (4)	0	0.12
**First Treatment**				0.003
Surgery	6 (19)	13 (46)	2 (10)	
RT	5 (16)	7 (25)	3 (15)	
Surgery + RT	1 (3)	4 (14)	4 (20)	
Surgery + RTCT	5 (16)	0	5 (25)	
RTCT	15 (47)	4 (14)	6 (30)	

**Table 3 jcm-14-07406-t003:** Delays and recurrence circumstances. The time to recurrence and the time between MTM and treatment are expressed as medians. (Q1; Q3).

	2019 N = 32 (100%)	2020 N = 28 (100%)	2021 N = 20 (100%)	*p*-Value
Time to recurrence (month)	22 (9; 46)	63 (25; 131)	48 (30; 102)	<0.001
Time between MTM and treatment (days).	30 (21; 45)	16 (13; 25)	24 (17; 27)	0.002
Circumstances of discovery *:				0.23
Clinical monitoring	16 (50)	8 (29)	5 (25)	
Scheduled radiology	5 (16)	4 (14)	2 (10)	
Patient symptoms	11 (34)	15 (54)	13 (65)	

* For one patient in 2020, this was an incidental finding on a CT scan performed for another reason (chest scan for suspected COVID-19 infection).

**Table 4 jcm-14-07406-t004:** Treatment performed after diagnosis of recurrence.

	2019 N = 32 (100%)	2020 N = 28 (100%)	2021 N = 20 (100%)
**Surgery**	7 (22)	9 (32)	4 (20)
**Radiotherapy**	1 (3)	2 (7)	2 (10)
**Radiochemoterapy**	2 (6)	3 (11)	2 (10)
**Surgery and adjuvant Radiotherapy**	3 (9)	7 (25)	3 (15)
**Surgery and adjuvant Radiochemotherapy**	8 (25)	3 (11)	0
**Exclusive Chemoterapy**	9 (28)	1 (4)	6 (30)
**No treatment realised**	2 (6)	3 (11)	3 (15)

**Table 5 jcm-14-07406-t005:** Characteristics of populations presenting a mismatch and a match between the treatment decided upon and the treatment carried out.

Characteristic	Missmatch MTM N = 19 ^1^	Match MTM N = 61 ^1^	*p*-Value ^2^
Age	66 (60, 71)	69 (59, 77)	0.5
Gender (men)	15 (79%)	53 (87%)	0.5
PS ≥ 2	7 (37%)	11 (18%)	0.12
COPD	3 (16%)	13 (21%)	0.7
Hepatopathy	1 (5.3%)	4 (6.6%)	>0.9
Cardiovascular disease	9 (47%)	36 (59%)	0.4
Diabetes	4 (21%)	16 (26%)	0.8
Undernutrition	15 (79%)	38 (62%)	0.3
Previous cancer history	5 (26%)	15 (25%)	>0.9
Active smoking	13 (68%)	45 (75%)	0.6
Alcoholism	9 (47%)	19 (32%)	0.3
Discovery of synchronous cancer	3 (16%)	4 (6.6%)	0.3
T > 2	11 (58%)	30 (49%)	0.6
N+	5 (26%)	15 (25%)	>0.9
M+	0 (0%)	8 (13%)	0.2
Time between MTM and treatment (days)	23 (14, 30)	24 (15, 38)	0.6
Time between end of first treatment and received diagnosis	76 (12, 131)	38 (18, 66)	0.4
3 years survival	3 (17%)	24 (39%)	0.094
Year			0.4
2019	6 (32%)	26 (43%)	
2020	6 (32%)	22 (36%)	
2021	7 (37%)	13 (21%)	

^1^ Median (Q1, Q3); n (%). ^2^ Wilcoxon rank sum test; Fisher’s exact test.

**Table 6 jcm-14-07406-t006:** Multivariate analysis of population factors that may influence the discrepancy between the treatment decided upon in MTM and the treatment actually provided.

Characteristic	OR	95% CI ^1^	*p*-Value
PS ≥ 2	1.17	0.92, 1.49	0.2
Undernutrition	1.01	0.82, 1.26	>0.9
Alcoholism	1.15	0.94, 1.41	0.2
Discovery of synchronous cancer	1.36	0.95, 1.96	0.10
T > 2	1.05	0.87, 1.27	0.6
N+	1.04	0.83, 1.29	0.7
M+	0.70	0.48, 1.01	0.060

^1^ CI = Confidence Interval.

## Data Availability

All data from this article are available upon request by contacting the corresponding author via e-mail: caroline.guigou@chu-dijon.fr.

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
