# Peer review of "Recurrence of Head and Neck Squamous Cell Carcinoma: Did the COVID-19 Pandemic Have an Impact on Therapeutic Management?"

_jcm, 2025, doi:10.3390/jcm14207406_

Round 1

Reviewer 1 Report

Comments and Suggestions for Authors

the authours evaluated the impact of covid on recurrent metastatic HNSCC

tables are clear and selection phase was properly described

although the topic is not new the finding is interesting. I believe that the authors should emphasize the role of MTD in the R/M HNSCC .

They observed discordance rate in 19% compared to between 5% and 23% for the population 
of newly diagnosed patients in the literature. No patient or tumor characteristics 
were identified that could explain this mismatch. In cases of treatment mismatch, the 
majority of patients received palliative care or no treatment at all (n=11, 59%). 

I believe this behavior is not influenced by COVID-19, but by the low interest of ENT surgeons and other physicians in the MDT in medical treatment. Indeed, the characteristics of the population with tumor recurrence were identical between 
the 3 periods. 

Author Response

Dear reviewer,

Thank you very much for your feedback and your work. We have added a paragraph to the discussion section stating that systemic treatment plays an important role and must be taken into account when making treatment decisions.

Reviewer 2 Report

Comments and Suggestions for Authors

I have read with great interest article entitled: Recurrence of Head and Neck Squamous Cell Carcinoma: Did the COVID-19 pandemic have an impact on therapeutic management? Based on the obtained results no factors were identified that could explain the discrepancy between the treatment decided upon and the treatment administered. The study population was homogeneous over the three years, with the only difference being lower lymph node staging in 2020. Also, the time to recurrence was longer (p< 0.001), and the time to treatment initiation was shorter during the pandemic and post-pandemic periods (p=0.002). This is interesting article dealing with availability of care for H&N cancer patients during COVID 19 pandemic. However, these results are in contrast to the most cancer series during COVID pandemic showing that time to treatment initiation was longer compared to pre/post-pandemic time. Also, in most series patients presented with more advanced stage (both T and N stage). Please comment this and propose potential explanation for such findings. Also, please provide Kaplan-Meier survival curves for different periods (pandemic, post-pandemic). Do you have information on radiotherapy initiation and duration during COVID pandemic compared to post-pandemic period?

Author Response

Dear reviewer,

The authors of this paper would like to thank you sincerely for your valuable comments and suggestions. We hope that our corrections will meet with your satisfaction. We remain at your disposal for any further modifications that may be necessary.

Best regards.

Q.1 Also, the time to recurrence was longer (p< 0.001), and the time to treatment initiation was shorter during the pandemic and post-pandemic periods (p=0.002). This is interesting article dealing with availability of care for H&N cancer patients during COVID 19 pandemic. However, these results are in contrast to the most cancer series during COVID pandemic showing that time to treatment initiation was longer compared to pre/post-pandemic time. Also, in most series patients presented with more advanced stage (both T and N stage). Please comment this and propose potential explanation for such findings.

A.1 Thank you very much for your comment. We were also surprised by these results concerning recurrence and treatment delays. We had previously conducted an initial study on the impact of the COVID-19 pandemic on the adequacy of treatments decided upon and treatments received, as well as on treatment delays for newly diagnosed patients (Reliquet et al., 2024). We also found shorter treatment delays.

We interpreted this as an indication of the French healthcare system's ability to adapt quickly to a pandemic. It would be interesting to conduct a national multicentre analysis to determine whether this finding is consistent across France or only in our centre.

This information is in the discussion (line 327): “However, there was a change in lymph node staging, with fewer locally advanced tumors detected during the pandemic and post-pandemic periods (p=0.002). Tumor recurrence times were longer (p=0.002) with a delay in treatment during the pandemic and post-pandemic periods (p<0.001). This finding aligns with our previous study [21], which demonstrated a shortened time to treatment once patients entered the healthcare system, thereby indicating the system's adaptive capacity and sustained functionality during the pandemic.”

Additional information has been added to the discussion concerning future work (line 375): “Nevertheless, it would be interesting to conduct a national multicenter analysis in order to compare the results of different French centers, in particular, response times during the pandemic. In this study, treatment initiation times were examined regardless of the treatment delivered. It would be interesting to examine treatment initiation times according to the type of treatment (surgery, radiotherapy or systemic treatment).”

Q.2 Also, please provide Kaplan-Meier survival curves for different periods (pandemic, post-pandemic).

A.2 We thank you for your suggestion to include Kaplan-Meier survival curves. However, in the context of our study, this approach is not the most appropriate. Indeed, the main objective was to evaluate the impact of the COVID-19 pandemic on the adequacy between treatments decided in MTM and those carried out. Patient survival was not one of the secondary objectives. Furthermore, there were no changes in patient survival over the 3-year follow-up period (Table 1). We therefore do not see any real benefit in including these curves in this study. We hope that our response is satisfactory.

Q.3 Do you have information on radiotherapy initiation and duration during COVID pandemic compared to post-pandemic period?

A.3 Sorry, in this study we did not record the duration of radiotherapy treatment. We also did not study differences in treatment initiation delays based on the treatments decided upon. This is another limitation of our work.

This sentence was therefore added to the discussion: “In this study, treatment initiation times were examined regardless of the treatment delivered. It would be interesting to examine treatment initiation times according to the type of treatment (surgery, radiotherapy or systemic treatment).”

Reviewer 3 Report

Comments and Suggestions for Authors

The manuscript entitled “Recurrence of Head and Neck Squamous Cell Carcinoma: Did the COVID-19 pandemic have an impact on therapeutic management?” presents a well-designed, comprehensive, and clinically meaningful study addressing a highly relevant and underexplored topic. The authors evaluate the impact of the COVID-19 pandemic on the adequacy between multidisciplinary team meeting (MTM) decisions and actual treatments performed in patients with recurrent head and neck squamous cell carcinoma (HNSCC).

This paper is methodologically solid, clearly structured, and particularly original, as it focuses on patients with tumor recurrence, a population that has been largely neglected in previous pandemic-related research. Most published studies have assessed the effects of COVID-19 on newly diagnosed head and neck cancers, whereas this work uniquely explores the real-world management of recurrent HNSCC, thus filling an important gap in the literature. The authors also place their findings in the context of existing national and international data, providing a well-integrated discussion and a nuanced interpretation.

The originality and clinical relevance of this manuscript cannot be overstated. It not only offers insight into how oncologic multidisciplinary teams adapted their decision-making processes during an unprecedented global health crisis, but also demonstrates the resilience and adaptability of the healthcare system. The observation that the concordance between MTM recommendations and treatments performed remained stable despite the pandemic is of particular value, as it suggests that high-quality oncologic decision-making and care continuity were successfully maintained even under extraordinary constraints. Similar adaptive trends have been described in other oncology settings during COVID-19, confirming that multidisciplinary coordination was preserved despite systemic pressure (please read and cite, doi: 10.1080/14740338.2022.1993819).

The manuscript is well written and logically organized. Figures and tables are informative and improve the overall readability of the text. The statistical analysis is sound, and the presentation of the data is transparent and detailed.

That said, a few minor improvements could further enhance the clarity and impact of the paper:

  1. Abstract: Consider emphasizing the novelty of the study population (patients with recurrent HNSCC), as this is the key element that differentiates this work from previous literature on the topic. Explicitly state that no previous study, to the authors’ knowledge, has specifically analyzed the impact of COVID-19 on the management of recurrent cases.

  2. Introduction: The introduction is informative and well contextualized. However, a brief paragraph could be added at the end to explicitly highlight the gap in knowledge and justify the originality of the present investigation.

  3. Methods: The methodology is clearly presented. Please confirm whether all patients provided informed consent specifically for inclusion in the registry or whether consent was waived under the CNIL MR-004 framework; the text already suggests both, but clarification would be welcome. In Table 1, consider harmonizing percentage formats (some are in parentheses, others not) and check minor alignment issues for improved readability.

  4. Results: The results are clearly described. Adding the total proportion of mismatched cases (n=19, 24%) in the first lines of section 3.4 would immediately orient the reader. The authors might briefly discuss the non-significant but numerically higher mismatch rate in 2021 (35%)—possibly as a delayed effect of healthcare reorganization or patient factors.

  5. Discussion: The discussion is well-balanced and comprehensive. It could be strengthened by emphasizing the original contribution of the study in providing reassurance that MTM decision integrity was preserved even for complex, recurrent cases—a particularly vulnerable group during the pandemic. It would be valuable to briefly compare the 24% mismatch rate observed here with those reported in pre-COVID literature (e.g., 15–25% range) to underscore that the pandemic did not exacerbate discrepancies. The paragraph on multidisciplinary decision-making could include a concise reflection on potential future improvements, such as digital/virtual MTM platforms or patient participation strategies, which the authors already mention but could elaborate slightly more.

  6. Limitations: The authors correctly acknowledge the retrospective and single-center design. They may wish to add a short note on generalizability: given the well-structured French MTM system, the results might be more robust than in other healthcare settings.

  7. Language: The English language is excellent. However, a few sentences could be streamlined for conciseness. For instance, “The observed discordance rate was 19% compared to between 5% and 23% for the population of newly diagnosed patients” could be rephrased as “The observed discordance rate (19%) was within the range reported for newly diagnosed populations (5–23%).”

Author Response

Dear reviewer,

The authors of this paper would like to thank you sincerely for your valuable comments and suggestions. We hope that our corrections will meet with your satisfaction. We remain at your disposal for any further modifications that may be necessary.

Best regards.

Q1. Abstract: Consider emphasizing the novelty of the study population (patients with recurrent HNSCC), as this is the key element that differentiates this work from previous literature on the topic. Explicitly state that no previous study, to the authors’ knowledge, has specifically analyzed the impact of COVID-19 on the management of recurrent cases.

A.1 Thank you. Now, this has been added into the abstract: “To our knowledge, no previous study has specifically investigated the impact of COVID-19 on the management of recurrent squamous cell carcinoma (HNSCC) of the upper aerodigestive tract.”

Q.2 Introduction: The introduction is informative and well contextualized. However, a brief paragraph could be added at the end to explicitly highlight the gap in knowledge and justify the originality of the present investigation.

A.2 Now this paragraph has been rewritten as follows (line 88): “To our knowledge, studies in the scientific literature dealing with this subject have focused only on newly diagnosed patients [21,24,25]. No previous study has specifically investigated the impact of COVID-19 on the management of recurrent cases. Patients with tumor recurrence have specific characteristics that do not allow the results for newly diagnosed patients to be extrapolated [10]. We therefore felt it was important to look into this subject, which is rarely addressed in scientific literature.”

Q.3 Methods: The methodology is clearly presented. Please confirm whether all patients provided informed consent specifically for inclusion in the registry or whether consent was waived under the CNIL MR-004 framework; the text already suggests both, but clarification would be welcome.

A.3 Thank you for your comment. Indeed, a letter of non-objection was sent to all patients whose medical records we used.

This sentence has now been added to the paper (line 110): “All patients received a letter informing them of the collection of data from their medical records and requesting their non-objection”.

Q.4 In Table 1, consider harmonizing percentage formats (some are in parentheses, others not) and check minor alignment issues for improved readability.

A.4 Table 1 has been revised. The figures in brackets correspond to percentages. I hope this suits you better.

Q.5 Results: The results are clearly described. Adding the total proportion of mismatched cases (n=19, 24%) in the first lines of section 3.4 would immediately orient the reader.

A.5 Now, this sentence has been changed to (line 273): “In our entire population, 19 patients (24%) received treatment different from that decided in the multidisciplinary team meeting (proportion of mismatched cases): 6 patients (19%) in 2019, 6 patients (21%) in 2020, and 7 patients (35%) in 2021. This difference is not significant between years (p=0.4, Wilcoxon test).”

Q.6 The authors might briefly discuss the non-significant but numerically higher mismatch rate in 2021 (35%)—possibly as a delayed effect of healthcare reorganization or patient factors.

A.6 Thank you for this remark. Now, this sentence has been added into the discussion (line 318): “However, the non-significant but numerically higher non-concordance rate in 2021 (35%) could be a delayed effect of healthcare reorganisation or patient-related factors.”

Q.7 Discussion: The discussion is well-balanced and comprehensive. It could be strengthened by emphasizing the original contribution of the study in providing reassurance that MTM decision integrity was preserved even for complex, recurrent cases—a particularly vulnerable group during the pandemic. It would be valuable to briefly compare the 24% mismatch rate observed here with those reported in pre-COVID literature (e.g., 15–25% range) to underscore that the pandemic did not exacerbate discrepancies.

A.7 Now the first paragraph of the discussion has been rewritten as follow (line 311): “This study aimed to investigate the impact of the COVID-19 pandemic on the management of recurrent head and neck cancer. Specifically, we assessed the concordance between MTM recommendations and the treatment actually administered, and identified factors potentially influencing any observed discordance. Our findings revealed that no increase in treatment mismatch was observed in patients with recurrent HSNCC due to the COVID-19 pandemic. The observed discordance rate (19%) was within the range reported for newly diagnosed populations (5–23%) [21–24]. No patient or tumor characteristics were identified that could explain this mismatch. However, the non-significant but numerically higher non-concordance rate in 2021 (35%) could be a delayed effect of healthcare reorganisation or patient-related factors. In cases of treatment mismatch, the majority of patients received palliative care or no treatment at all (n=11, 59%). The integrity of decisions regarding therapeutic drug management was preserved, even in complex and recurrent cases, a particularly vulnerable group during the pandemic. These results demonstrate the value of systemic treatments for this patient population. It is important to consider this during multidisciplinary team meetings.”

Q.8 The paragraph on multidisciplinary decision-making could include a concise reflection on potential future improvements, such as digital/virtual MTM platforms or patient participation strategies, which the authors already mention but could elaborate slightly more.

A.8 Now, this paragraph of the discussion has been rewritten as follow (line 343): “Our study is part of a study of MTM practice, which remains an essential and mandatory routine in medical and surgical oncology. Beyond the legal framework, it allows for the comparison of perspectives from different but complementary medical specialties [37]. A persistent challenge, however, lies in the final treatment decision, which necessitates shared decision-making with the patient. MTMs must be improved to take into account patients as a whole. In our institution, patients do not directly participate in MTM meetings. This may be an area for improvement, even though a recent national study in our specialty showed no difference [38]. Digital or virtual MTMs could also be developed to enable remote patient participation. The development of artificial intelligence could also improve MTMs decisions in the coming years by comparing them to the guidelines issued by learned societies.”

Q.9 Limitations: The authors correctly acknowledge the retrospective and single-center design. They may wish to add a short note on generalizability: given the well-structured French MTM system, the results might be more robust than in other healthcare settings.

A.9 Now, this sentence has been added into the discussion (line 383): “With regard to the generalizability of the results, it is important to note the well-structured French MTM system. The results might be more robust than in other healthcare settings.”

Q.10 Language: The English language is excellent. However, a few sentences could be streamlined for conciseness. For instance, “The observed discordance rate was 19% compared to between 5% and 23% for the population of newly diagnosed patients” could be rephrased as “The observed discordance rate (19%) was within the range reported for newly diagnosed populations (5–23%).”

A.10 Now, this sentence: “The observed discordance rate was 19% compared to between 5% and 23% for the population of newly diagnosed patients” has been changed by “The observed discordance rate (19%) was within the range reported for newly diagnosed populations (5–23%).”

Some sentences in English have also been reworded in the text.